# Crystallographically Textured and Magnetic LaCu-Substituted Ba-Hexaferrite with Excellent Gyromagnetic Properties

**DOI:** 10.3390/ma15248792

**Published:** 2022-12-09

**Authors:** Zan Jiao, Yuantao Wang, Meng Wei, Qifan Li, Ziyu Li, Alexander S. Sokolov, Chengju Yu, Xiaona Jiang, Chuanjian Wu, Zhongwen Lan, Ke Sun, Zhong Yu, Vincent G. Harris

**Affiliations:** 1School of Materials and Energy, University of Electronic Science and Technology of China, Chengdu 610054, China; 2Chengdu Liaoyuan Xingguang Electronics Company Limited, Chengdu 610100, China; 3Center for Microwave Magnetic Materials and Integrated Circuits (CM3IC), Department of Electrical and Computer Engineering, Northeastern University, Boston, MA 02115, USA

**Keywords:** Ba-hexaferrite, remanent magnetization, ferromagnetic resonance (FMR) linewidths, anisotropy fields, effective linewidth

## Abstract

Excellent gyromagnetic properties of textured, bulk Ba-hexaferrite samples are required for low-loss, self-biased applications for microwave and millimeter-wave (MMW) devices. However, conventionally processed bulk Ba-hexaferrite ceramics typically demonstrate low remanent magnetization values, 4π*M*_r_, of 2.0~3.0 kG, and relatively large ferromagnetic resonance (FMR) linewidths, Δ*H_FMR_*, of 0.8~2 kOe. These properties lead to the development of high-performance, practical devices. Herein, crystallographically textured Ba-hexaferrite samples, of the composition Ba_0.8_La_0.2_Fe_11.8_Cu_0.2_O_19_, having excellent functional properties, are proposed. These materials exhibit strong anisotropy fields, *H*_a_, of ~14.6 kOe, high remanent magnetization, 4π*M*_r_, of 3.96 kGs, and a low Δ*H_FMR_* of 401 Oe at zero-bias field at the Q-band. Concomitantly, the broadband millimeter-wave transmittance was utilized to determine the complex permeability, *μ**, and permittivity, *ε**, of textured hexaferrites. Based on Schlöemann’s theory of complex permeability, *μ**, the remanent magnetization, 4π*M*_r_, anisotropy field, *H*_a_, and effective linewidth, Δ*H_eff_*, were estimated; these values agree well with measured values.

## 1. Introduction

Ferrite materials provide the necessary magnetic media to break time-reversal symmetry, allowing non-reciprocal behavior to passive control elements, such as circulators and isolators [1,2,3]. Despite progress over the past several decades, these devices remain comparatively large and heavy, mostly due to the external permanent magnets required to provide the necessary bias fields to saturate the ferrite. On the other hand, hexaferrites have a sufficiently high crystallographic texture that permits the exploitation of their magnetocrystalline anisotropy fields, *H_a_*, affording them the potential for miniaturization and improved integration of microwave and millimeter-wave (MMW) devices. Specifically, M-type Ba-hexaferrites (i.e., BaFe_12_O_19_, BaM) have attracted much attention for applications in monolithic-microwave-integrated circuits (MMIC) due to their strong and adjustable *H*_a_, that acts as an effective internal field [4,5,6,7].

Broadband, low-loss properties are prerequisites for high-performing, high-frequency, and self-biased MMW devices based on crystallographically textured Ba-hexaferrites [3]. Improved remanent magnetization, 4π*M_r_*, and narrow ferromagnetic resonance linewidths, Δ*H_FMR_*, allow for the reduction in resonance losses [8] and optimized anisotropy fields, *H_a_*. These properties allow for the RF engineer to design the operational frequency (i.e., center) and device performance based on material properties. Figure 1a illustrates published values of remanent magnetization, 4π*M_r_*, and FMR linewidth, Δ*H_FMR_*, of bulk Ba-hexaferrites [9,10,11,12,13,14,15,16,17]. It is clear that conventionally processed bulk Ba-hexaferrites usually exhibit a relatively low 4π*M*_r_ of 1.5~3.5 kGs and broad Δ*H_FMR_* of 0.5 to 2 kOe. Herein, we demonstrate crystallographically textured hexaferrites with concomitant high 4π*M_r_* and low Δ*H_FMR_*.

Furthermore, dynamic magnetic permeability, *μ**, and dielectric permittivity, *ε**, are two fundamental parameters that determine the response of ferrites to high-frequency electromagnetic fields. We employed high-*Q* resonant measurements to determine dielectric and magnetic properties at microwave frequencies [14]. The accuracy of this method declines in the millimeter-wave region of the electromagnetic spectrum. Therefore, the magneto-optical approach has been employed over this important frequency range. This technique allows for the separation of dielectric and magnetic effects in ferrites enabling the precise characterization of the dispersive properties of ferrites over the entire millimeter-wave range [18].

Basic operational principles of MMW devices require the thickness of crystallographically textured ferrites often exceeding 100 μm [6], making bulk hexaferrites more preferable to their thin-film counterparts. In the present work, we investigate and report on the processing–structure–property relationships of La-Cu-substituted Ba-hexaferrites prepared by conventional ceramic processes. The crystalline texture, microstructure, magnetic, and microwave properties have been investigated. The complex permeability, *μ**, complex permittivity, *ε**, and effective linewidth, Δ*H*_eff_, are determined by the magneto-optical measurements.

## 2. Experimental Procedures

Previous calculations [19] of density functional theory (DFT) on similar hexaferrites revealed that La^3+^ ions are mainly substituted Ba^2+^ ions, and Cu^2+^ ions reside on the 2*b* and 4*f*_2_ sites (see Figure 1b). Introduction of the Cu^2+^ ions allow for adjustment of microwave and magnetic properties of crystallographically textured Ba-hexaferrites. Here, Ba_0.8_La_0.2_Fe_11.8_Cu_0.2_O_19_ materials were synthesized according to conventional ceramic processes. Specifically, analytical-grade raw materials of BaCO_3_, Fe_2_O_3_, La_2_O_3_, and CuO powders were homogeneously mixed by ball-milling in an agate jar for 12 h. These composite mixtures were subsequently calcined from 1100 to 1200 °C for 2 h, doped with 2.5 wt.% of Bi_2_O_3_ and 2.0 wt.% of CuO, and ball-milled again for 18 h to achieve a mean particle size of 0.6~1.0 μm. Following the ball-milling, the powders were dried, and then deionized water was added to form an aqueous-based slurry with a 60~70 wt.% solids loading. Subsequently, the slurries were uniaxially pressed in a longitudinal magnetic field (i.e., ~15 kOe) to produce a crystallographically textured Ba-hexaferrite cylindrical green compact, with the *c*-axes of Ba-hexaferrites particles aligned mostly perpendicular to the plane of the cylinder. The specimens were then sintered at 950~1050 °C for 2 h.

The crystallographic structure was determined by X-ray diffraction (XRD, Bruker-AXS D8 advance, Karlsruhe, Germany) using a Cu-*K*_α_ radiation source. The cross-section morphology was observed using a field emission scanning electron microscope (FESEM, JEOL JSM-7800F, JEOL, Tokyo, Japan). The bulk density, *d*, was measured using a Micromeritics AccuPyc 1330 pycnometer. Room-temperature magnetic properties were determined by vibrating sample magnetometry (VSM, Quantum Design SQUID, Quantum Design Inc., San Diego, CA, USA) with a maximum applied magnetic field of 20 kOe. The absorbed power spectra were characterized by the TE_10_ rectangular transmission cavity perturbation at Q-band (30~50 GHz) [20,21]. Broadband millimeter-wave measurements were performed using a free-space quasi-optical spectrometer equipped with a series of backward wave oscillators (BWOs) as high-power, tunable sources of coherent radiation over a frequency range of 30~90 GHz.

## 3. Results and Discussion

Figure 2a shows an X-ray diffraction pattern collected at room temperature from crystallographically textured Ba-hexaferrite powder samples. All diffraction peaks have been indexed according to the standard powder diffraction pattern of the JCPDF card (No.43-0002) for Ba-hexaferrites, corresponding to space group *P*6_3_/*mmc*. The peaks perpendicular to the sample plane, corresponding to the c-axes reflections of (006), (008), and (0014), have the highest intensity. However, other weaker peaks are also present, namely (107), (206), and (220). As a reliable way to evaluate the degree of crystalline orientation, the Lotgering factor, *f*_L_, was calculated [22]
(1)$$f_{L}=\frac{\sum I(00l)/\sum I(hkl)-\sum I_{0}(00l)/\sum I_{0}(hkl)}{1-\sum I_{0}(00l)/\sum I_{0}(hkl)}$$
where ∑*I*(00*l*)/∑*I*(*hkl*) is calculated based on the experimental data for crystallographically textured samples, and *∑I*_0_(00*l*)*/∑I*_0_(*hkl*) is based on the XRD data from the JCPDF card. The evaluated degree of crystalline orientation, *f*_L_, for Ba_0.8_La_0.2_Fe_11.8_Cu_0.2_O_19_ samples reaches 83%, indicating the degree of the *c*-axis-preferred orientation.

The cross-sectional morphology of crystallographically textured Ba-hexaferrites is presented in Figure 2b. It can be observed that the textured Ba_0.8_La_0.2_Fe_11.8_Cu_0.2_O_19_ sample demonstrates a microstructure consisting of hexagonal particles compactly aligned parallel to the *c*-axis. It is crucial to control the grain size during processing of textured samples in order to attain high remanent magnetization, 4π*M_r_*. We measure grain sizes to range from 1 to 2 μm, with an average grain size of 1.2 μm. This value lies within the estimated single domain critical size range of 1.0–1.5 μm [23]. Additionally, the density, *d*, of the as-sintered Ba_0.8_La_0.2_Fe_11.8_Cu_0.2_O_19_ specimen was measured to be 97.5% of the theoretical density, ~5.28 g/cm^3^ [24]. As indicated previously in Refs. [25,26], bismuth copper oxides, generated in the reaction of Bi_2_O_3_ and CuO, establish a liquid channel at the grain boundaries. It is speculated that this structure promotes densification and grain alignment during sintering. Yet, some pores are still visible in Figure 2b after sintering, and these inevitably contribute to the broadening of the FMR linewidth.

Figure 2c illustrates representative static magnetic hysteresis loops (*M* versus *H*) of a crystallographic and magnetically textured Ba_0.8_La_0.2_Fe_11.8_Cu_0.2_O_19_ sample. The kink in the out-of-plane loop evolves slowly, thus indicating the strong *c*-axis orientation in the textured Ba_0.8_La_0.2_Fe_11.8_Cu_0.2_O_19_ sample. Because of a preferred orientation grain in the sample, the hysteresis loops between the in-plane and out-of-plane loops are obviously different. The reason can be explained that the magnetization prefers the direction of the *c*-axis. Thus, the magnetization of the sample with an external field applied in *c*-axis direction is larger than perpendicular to *c*-axis under same magnetic field. To describe obvious divergences between the in-plane and out-of-plane loops, the anisotropy field, *H_a_*, was obtained by the singular point detection (SPD) method [17], which was the abscissa corresponding to the maximum value in the second derivative (d^2^*M*/d*H*^2^) of initial magnetization curve measured perpendicular to the direction of easy magnetization [27]. Subsequently, the anisotropy constant *K*_1_ was calculated from the basic equation *H_a_* = 2*K*_1_/*M*_s_. Representative magnetic properties are summarized in Table 1. It was found that textured Ba_0.8_La_0.2_Fe_11.8_Cu_0.2_O_19_ samples show a strong uniaxial magnetic anisotropy with *H_a_* values of ~14.6 kOe and anisotropy constant *K*_1_ values of 2.52 × 10^5^ J/m^3^. The measured 4π*M*_r_ of 3.96 kGs is superior to previously reported results for bulk BaM (see Figure 1a). This improvement in 4π*M_r_* is mainly due to uniformity in crystalline grains stemming from the higher sintered density, compared with Ref. [25]. Such a strong anisotropy, *H_a_*, and high remanent magnetization, 4π*M*_r_, enable MMW devices to operate without bias magnets [6,7,15].

The frequency response of absorbed power at the Q-band was measured with external biasing fields, *H,* ranging from 0 to 1 kOe, applied parallel to the cylinder axes. As depicted in Figure 3, the absorbed power spectra were fitted to a Lorentzian function consisting of symmetric and asymmetric components [28]:(2)$$P=L_{sym}\frac{\Delta f^{2}}{{\left(f-f_{res}\right)}^{2}+\Delta f^{2}}+D_{asym}\frac{\Delta f(f-f_{res})}{{\left(f-f_{res}\right)}^{2}+\Delta f^{2}}$$
where Δ*f* denotes the half width at half maximum (HWHM), and *f_res_* is the ferromagnetic resonance frequency. *L_sym_* and *D_asym_* are symmetric and asymmetric contributions to the frequency, respectively. The corresponding Δ*H_FMR_* of the textured Ba_0.8_La_0.2_Fe_11.8_Cu_0.2_O_19_ sample was evaluated according to the equation Δ*H_FMR_* = 2 × Δ*f*/*γ* and listed in Table 1. The strong absorption zones reside in the vicinity of the respective center frequencies of 44.0 GHz and 46.3 GHz for crystallographically textured Ba-hexaferrites at *H* = 0 and 1 kOe, respectively. According to the Kittel formula [17], the anisotropy field, *H_a_*, obtained from FMR measurements, was extrapolated to be 15.1 ± 0.2 kOe, which is in close agreement with the value measured from the SPD method, 14.6 kOe. It is noteworthy that the crystallographically textured Ba_0.8_La_0.2_Fe_11.8_Cu_0.2_O_19_ sample exhibited a narrow Δ*H_FMR_* of 401 Oe at zero-bias field, an improvement of over 20% compared to the previous values for bulk BaM (see Figure 1a).

In order to probe the sources of the FMR linewidth, Δ*H_FMR_*, in hope of decreasing it further, one must understand and estimate the contributions to the FMR linewidth. In polycrystalline ferrites, the total FMR linewidth, Δ*H_FMR_*, depends crucially on the superposition of intrinsic and extrinsic contributions [29].
(3)$$\Delta H_{FMR}=\Delta H_{i}+\Delta H_{a}+\Delta H_{p}$$
where Δ*H_i_* is the intrinsic linewidth; Karim et al. [30] speculated that Ba-hexaferrites possess an intrinsic linewidth of 0.3~0.4 Oe/GHz. Δ*H_a_* and Δ*H_p_* relate to the crystalline anisotropy and porosity-induced linewidth-broadening contributions. Approximate estimations of Δ*H_a_* and Δ*H_p_* were proposed by Schlömann [31,32], based on the independent grain approaches with large anisotropy (*H_a_* >> 4π*M*_s_):(4)$$\left\{ \begin{array}{l} \Delta H_{a}\approx 0.87H_{a}\\ \Delta H_{p}\approx 1.5\left(4\pi M_{s}\right)P \end{array}\right.$$
where *P* corresponds to porosity. This evaluation indicates 80~85% of the total FMR linewidth is determined by the porosity contribution. In other words, a relatively high density of 97.5% could allow for an even lower FMR linewidth in textured Ba_0.8_La_0.2_Fe_11.8_Cu_0.2_O_19_ samples.

The millimeter-wave transmission spectrum of textured Ba_0.8_La_0.2_Fe_11.8_Cu_0.2_O_19_ samples is presented as Figure 4. A relatively wide absorption region centered at about 44.3 GHz is believed to be a natural ferromagnetic resonance [33]. Nevertheless, the actual width of resonance could not be ascertained from the absorption profile, which is ascribed mainly to the broadening introduced by the saturation of the absorption line [18]. At frequencies below and above this region, a progressive decline of the transmissivity allows for oscillations to be observed. Such variations in the microwave or millimeter ranges are completely associated with magnetic permeability [34]. By integrating the relationship between the transmittance and reflectance spectra, and also refractive and absorption indexes, we obtain the following [35]:(5)$$\left\{ \begin{array}{l} T=E\frac{{\left(1-R\right)}^{2}}{{\left(1-RE\right)}^{2}+4RE\sin^{2}(2\pi n\nu d/c)}\\ R=\frac{{\left(n-1\right)}^{2}+k^{2}}{{\left(n+1\right)}^{2}+k^{2}}\\ E=\exp(-\frac{4\pi kdf}{c})\\ n+ik=\sqrt{\varepsilon^{*}\mu^{*}} \end{array}\right.$$
where *T*, *R*, *n*, *k*, *μ**, and *ε** are the transmittance, reflectance, refractive index, absorption index, complex permeability, and complex dielectric permittivity, respectively. The dielectric permittivity, *ε**, typically exhibits a very weak frequency dependence in the millimeter range, and therefore was reasonably assumed to be constant. By fitting the transmission spectrum far from the absorption region, it was deduced that the textured Ba_0.8_La_0.2_Fe_11.8_Cu_0.2_O_19_ sample had a real permittivity *ε*’ of 23, and an imaginary permittivity *ε*” of 0.29. The complex magnetic permeability was estimated based on the magnetocrystalline anisotropy, *H_a_*, and remanent magnetization, 4*πM_r_* [36]:(6)$$\mu^{*}/\mu_{0}=sqrt\{[{\left(f_{a}^{*}+f_{4\pi Mr}\right)}^{2}-f^{2}]/[{\left(f_{a}^{*}\right)}^{2}-f^{2}]\}$$
where *f_a_*^*^ = (*γ*/2π)*H_a_* + *jfG*, and *f*_4π*M*_*_r_* = 2*γ*(4π*M_r_*) with *γ* being the gyromagnetic ratio, and *G* being the damping parameter. The fitting of the experimentally observed spectrum was performed by using Equation (6). The significant discrepancy between the experimental and simulated results for the resonance absorption region is attributed to the considerable contribution of “non-intrinsic” relaxation. The best-fit dissipation leads to *H_a_* = 15.4 kOe, 4π*M_r_* = 4.02 kGs, and *G* = 0.0011 far from the resonance. These values of *H_a_* and 4π*M_r_* agree well with those obtained from VSM and FMR measurements. According to the equation Δ*H_eff_* = 2π*G·f*/*γ* [33], the effective linewidth, Δ*H_eff_*, equals 108 Oe. This narrow Δ*H_eff_* is beneficial for the design of MMW devices that operate far from the FMR frequency.

## 4. Conclusions

In summary, Ba_0.8_La_0.2_Fe_11.8_Cu_0.2_O_19_ ferrites having both strong crystallographic and magnetic texture were successfully fabricated by employing conventional ceramic processes. These ferrites exhibit a strong crystallographic *c*-axis alignment, high sintering density, *d,* of 5.15g/cm^3^, strong anisotropy field, *H_a_*, of 14.6 kOe, and remanent magnetization, 4π*M_r_*, values up to 3.96 kGs. The corresponding FMR linewidth, Δ*H_FMR_*, with biasing fields, *H*, were measured to range from 0 and 1 kOe at the Q-band to 401 Oe and 379 Oe, respectively. Moreover, magneto-optical measurements allowed for the determination of the complex dielectric permittivity, *ε**, permeability, *μ**, and effective linewidth, Δ*H_eff_*. While the permeability, *μ**, is strongly frequency-dependent, the real (*ε*’) and imaginary parts (*ε*”) of the dielectric permittivity, *ε**, and effective linewidth, Δ*H_eff_*, are 23, 0.29, and 108 Oe away from resonance, respectively. These results indicate that Ba_0.8_La_0.2_Fe_11.8_Cu_0.2_O_19_ ferrites demonstrating both strong crystallographic and magnetic texture could be employed in self-biased and low-loss MMW devices, such as circulators or isolators.

## Figures and Tables

**Figure 1 materials-15-08792-f001:**
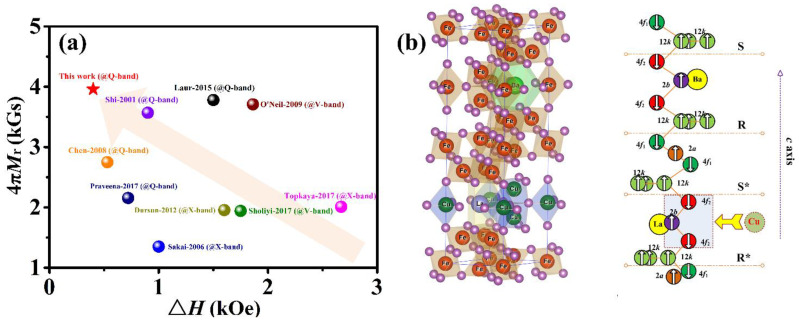
(**a**) Published values for 4π*M_r_* and Δ*H_FMR_* for bulk barium hexaferrite compositions relevant to the present work. The upper left quadrant is the region representing ideal properties. (**b**) Atomic unit cell with spin distribution for the La-Cu substituted barium hexaferrites studied here.

**Figure 2 materials-15-08792-f002:**
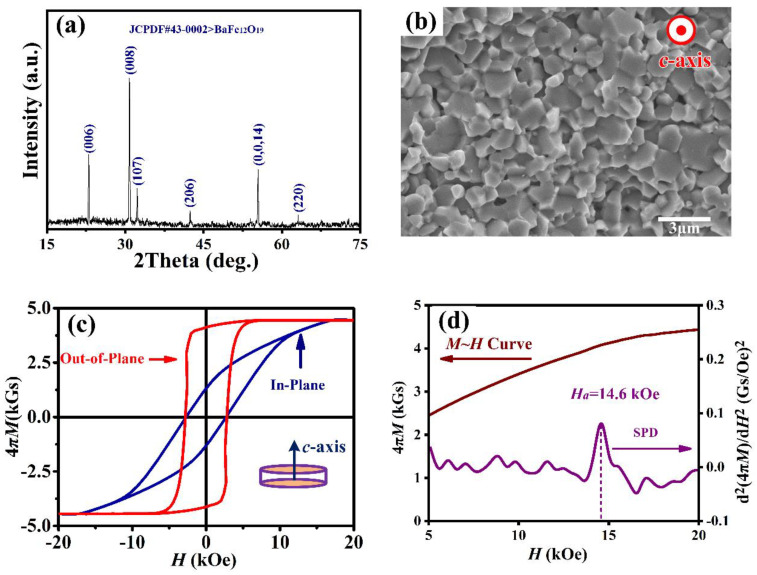
(**a**) X-ray diffraction pattern from Ba_0.8_La_0.2_Fe_11.8_Cu_0.2_O_19_ sample illustrating strong crystallographic texture. (**b**) Scanning electron microscopy image revealing a dense cross-sectional microstructure of faceted crystals. (**c**) Hysteresis loops collected with the applied magnetic field aligned perpendicular to and in-plane with the sample plane. (**d**) Initial magnetization curve and second derivative d^2^(4π*M*)/d*H*^2^ curve measured perpendicular to the direction of easy magnetization.

**Figure 3 materials-15-08792-f003:**
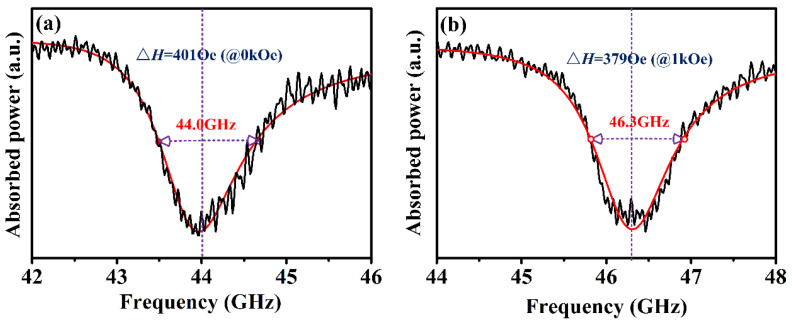
The dependence of the frequency response of absorbed power in a crystallographically textured Ba_0.8_La_0.2_Fe_11.8_Cu_0.2_O_19_ sample: (**a**) at 0 Oe; (**b**) at 1 kOe.

**Figure 4 materials-15-08792-f004:**
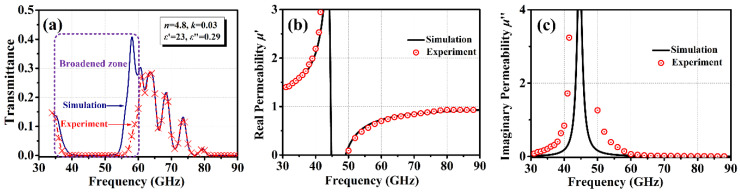
Millimeter-wave transmittance spectra, and real and imaginary parts of magnetic permeability in crystallographically textured Ba_0.8_La_0.2_Fe_11.8_Cu_0.2_O_19_ sample as a function of frequency: (**a**) Millimeter-wave transmittance spectra; (**b**) real parts of magnetic permeability; (**c**) imaginary parts of magnetic permeability.

**Table 1 materials-15-08792-t001:** Physical, magnetic, and gyromagnetic properties of textured barium hexaferrites. ⊥ and // denote the applied magnetic field perpendicular and parallel to hexagonal crystallographic c-axis.

*d* (g/cm^3^)	4π*M_s_* (kGs)	4π*M_r_* (kGs)	*H_c_* (Oe)	*H_a_* (kOe)	*K_1_* (×10^5^ J/m^3^)	Δ*H_FMR_* (Oe)
5.15	4.45	3.96 (⊥) 1.22 (//)	2813 (⊥) 2698 (//)	14.6	2.52	379^@1kOe^ 401^@0kOe^

## Data Availability

The data used to support the findings of this study are available from the corresponding author upon request.

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
