# Peer review of "Crystallographically Textured and Magnetic LaCu-Substituted Ba-Hexaferrite with Excellent Gyromagnetic Properties"

_materials, 2022, doi:10.3390/ma15248792_

Round 1
Reviewer 1 Report
The manuscript covers almost all important parameters like the calculated density (to roughly estimate porosity), and its contribution to FMR linewidth studies, shows relatively improved remanent magnetization which is essential for the device application. Maybe accepted for publication after these mandatory revisions.
1) The difference between the theoretical and calculated density gives a rough estimate of pores. In ref 17, the density (cal) was 92.5 % which is much less than the calculated value in this paper but still shows better line width measurements. Hence a more magnified SEM image would give a better perception of the pores
2) Further description of variation in the in-plane and out-of-plane magnetization would surely help the reader further (SPD method)
Author Response
We thank the editor and reviewers for the comments concerning our manuscript entitled “Crystallographic and magnetic textured LaCu-substituted Ba-hexaferrite with excellent gyromagnetic properties” (ID: materials-2024038). Those comments are all valuable and very helpful for improving and revising our paper, as well as the most important guiding significance to our research. We have studied the comments carefully and made corrections, which we hope to meet with approval. The response to the comments is presented below, and the corresponding changes are highlighted in red in the revised manuscript. Please see the attachment

Reviewer 2 Report
The present manuscript shows the excellent properties of LaCu-doped Ba-hexaferrite. The structural, microstructural, magnetic and the ferromagnetic resonance properties of the hexaferrites have been studies in detail. The results are interesting, and the manuscript is written in a clear and concise manner. I recommend publication of this manuscript without any further changes to the text. However, the resolution of the figure files of the manuscript needs to be significantly improved before publication for the convenience of the readers.
Author Response
We thank the editor and reviewers for the comments concerning our manuscript entitled “Crystallographic and magnetic textured LaCu-substituted Ba-hexaferrite with excellent gyromagnetic properties” (ID: materials-2024038). Those comments are all valuable and very helpful for improving and revising our paper, as well as the most important guiding significance to our research. We have studied the comments carefully and made corrections, which we hope to meet with approval. The response to the comments is presented below, and the corresponding changes are highlighted in red in the revised manuscript. Please see the attachment.

Reviewer 3 Report
This work reports on the improved gyromagnetic properties of M-type LaCu-substituted Ba-hexaferrites, and their application to MMW devices. The authors organized their paper systematically using supporting results such as XRD, FESEM, and M versus H, and provided a comprehensive understanding regarding the relation between the properties of the Ba-hexaferrite and its device application. However, there are some ambiguities to be addressed as follows.
- What are the exact roles of La and Cu in the Ba-hexaferrites? Please describe it in detail.
- There are no results for the control group. The authors claimed that the improvement in 4πMr is mainly due to uniformity in crystalline grains stemming from the higher sintered density. At least one different composition other than Ba0.8La0.2Fe11.8Cu0.2O19 or one other synthesis condition is required to perform additional experiments to support the improvement caused by the high density.
- The authors should describe how they determined the composition of the LaCu-substituted Ba-hexaferrite. Also, what is the concentration of bismuth in the ferrite?
- A description of the simulation method was omitted.
- The authors mentioned that it is crucial to control the grain size in order to attain high remanent magnetization. However, they only reported the result for an average grain size of 1.2 μm. Please provide the experimental results for other grain sizes or the detailed explanation of the grain size effect with some literatures.
Author Response

(The authors gave the same response as above.)

Round 2
Reviewer 1 Report
All the comments have been addressed satisfactorily. Maybe accepted.